# Valorization of Spent Brewer's Yeast Bioactive Components via an Optimized Ultrasonication Process

Livia Teodora Ciobanu [1,2], Diana Constantinescu-Aruxandei [1], Naomi Tritean [1,3], Carmen Lupu [1], Radian Nicolae Negrilă [4], Ileana Cornelia Farcasanu [2,*] and Florin Oancea [1,5,*]

[1]  Bioresources Department, National Institute for Research & Development in Chemistry and Petrochemistry—ICECHIM, Spl. Independentei No. 202, Sector 6, 060021 Bucharest, Romania; livia.ciobanu@icechim.ro (L.T.C.); diana.constantinescu@icechim.ro (D.C.-A.); naomi.tritean@icechim.ro (N.T.); carmen.lupu@icechim.ro (C.L.)
[2]  Interdisciplinary School of Doctoral Studies ISDS—UB, University of Bucharest, Bd. Mihail Kogalniceanu No. 36–46, 050107 Bucharest, Romania
[3]  Faculty of Biology, University of Bucharest, Splaiul Independentei No. 91–95, 050095 Bucharest, Romania
[4]  SC Agsira Srl, Str. Nicolae Bălcescu No. 54, Hala 2, 207340 Ișalnița, Dolj County, Romania; negrilar@yahoo.com
[5]  Faculty of Biotechnologies, University of Agronomic Sciences and Veterinary Medicine of Bucharest, Bd. Mărăști No. 59, Sector 1, 011464 Bucharest, Romania
*   Correspondence: ileana.farcasanu@chimie.unibuc.ro (I.C.F.); florin.oancea@icechim.ro (F.O.); Tel.: +40-21-316-3071 (F.O.)

**Abstract:** The increasing need for sustainable waste management and food fortification requires continuous agri-food biotechnological innovation. Spent brewer's yeast (SBY) is a mass-produced underutilized by-product of the brewery industry and has elevated bioactive potential. The current study presents a streamlined ultrasonic SBY cell lysis method, with the main goal of bioactive compound valorization. The influence of selected ultrasonication parameters on protein release and, implicitly, on the cell disruption efficiency, was assessed. The SBY derivatives resulting from the ultrasonic cell lysis were SBY extracts (SBYEs) and cell walls (SBYCWs), which were evaluated in terms of protein content, antioxidant activity (AOA) and total polyphenol content. Scanning electron microscopy (SEM) and FT-IR spectroscopy were used to characterize SBYCWs in relation to the morphological and chemical transformations that follow ultrasonic yeast cell disruption. The optimal ultrasonication conditions of 6.25% SBY concentration, 40 °C and 33.33% duty cycle (DC) ensured the most efficient lysis. The SBY derivatives with the most elevated antioxidant activity were obtained at temperatures below 60 °C. SBYCWs had the highest polyphenol content and a relatively high content of β-glucan under these parameters. Optical microscopy and SEM confirmed the release of intracellular content and separation of SBYCWs.

**Keywords:** extracts; cell walls; β-glucan; protein; polyphenols; antioxidant activity; optimized parameters





## 1. Introduction

The great impact of nutrition on health has been recognized since ancient times. Scientific research in the last decades has confirmed the importance of diet on overall well-being. Functional foods and ingredients, dietary supplements and agri-food biotechnologies are now receiving unprecedented attention in the continuous strive to improve the quality of life with regard to good health maintenance and disease prevention [1–5]. Microorganisms, including yeasts, are becoming a resourceful tool in innovating these food sectors [6–8]. Yeast-based foods and beverages are known for their health-promoting benefits, as these microorganisms produce high amounts of bioactive compounds with prebiotic, postbiotic, antioxidant, antihypertensive and immunomodulator effects [9–12]. Spent yeasts are among the main by-products of fermenting beverages and food industries. Some studies have revealed the nutraceutical value of these residues [13–15]. Valorizing the potential

of spent yeasts in food, cosmetic or pharmaceutical sectors could give rise to valuable products with therapeutic or prophylactic benefits while meeting the needs of sustainable management of industrial wastes.

Cell lysis or cell disruption is the process of disintegrating the plasma membrane, which results in the release of intracellular content. This technique represents a primary step in extracting and valorizing the bioactive compounds from spent yeast cells. Cell disruption is commonly used to obtain yeast derivatives with elevated nutritive potential, mainly yeast extracts and yeast cell walls [16–19].

Various mechanical methods (ultrasonication, bead milling or high-pressure homogenization) or non-mechanical methods (enzymatic, chemical or physical lysis) can be used to achieve cell disruption. Ultrasonication emerges as a feasible method for cell lysis. The process is simple, time-saving and highly effective in comparison to other disruption methods. Moreover, ultrasonication can be considered a mild lysis method, as it does not require any harsh procedures or reagents that might affect the extracted bioactive compounds. Depending on the processing conditions, the advantages of ultrasonication extend to its capacity to act as a potential flavor modulator. Thus, ultrasonication might enhance the taste of certain fermenting foods (e.g., ultrasounds can improve the enzymatic activity of proteases for flavor maturation acceleration in soy sauce) or contribute to debittering of some beverages (e.g., ultrasound could enhance enzymatic hydrolysis of bitter compounds in ougan juice) [20,21]. These applications could facilitate further uses of ultrasonicated products i.e., spent brewer's yeast derivatives, in the food industry. The principle of this method relies on the capacity of high-frequency ultrasounds to generate gas bubbles (cavities) within a liquid sample, which will eventually collapse, a phenomenon known as cavitation. The continuous collapse of the bubbles generates shock waves which will lead to the disruption of the plasma membrane [16,17,22,23].

Although spent brewer's yeast cells have significant nutraceutical value due to their ability to adsorb bioactive compounds produced from ingredients like hops during the brewery process, reports on spent brewer's yeast cell lysis are still scarce [13]. The current study presents a streamlined ultrasonication method for spent brewer's yeast cell lysis. The influences of the main parameters, such as sample concentration, temperature and duty cycle were assessed in relation to the disruption efficiency, protein content and antioxidant activity. The yeast cell walls were also characterized and the Pearson correlation between all responses was analyzed. This study also highlights the nutritive potential of spent brewer's yeast derivatives, specifically spent brewer's yeast extracts and spent brewer's yeast cell walls, with regard to their protein and polyphenol content, respectively, and antioxidant activity. The yeast cell walls obtained from spent brewer's yeast were additionally characterized using scanning electron microscopy, β-glucan content and FT-IR assay.

## 2. Materials and Methods

### 2.1. Materials

Spent brewer's yeast (SBY), *Saccharomyces pastorianus*, was kindly provided as a slurry by AGSIRA SRL from a local brewery in Romania. The following reagents were used: bovine serum albumin (BSA) Fraction V, NZ-Origin (Carl Roth Gmbh, Karlsruhe, Germany), copper sulfate crystals, potassium iodide, potassium ferricyanide (Reactivul București, Bucharest, Romania), sodium potassium tartrate, sodium hydroxide pellets, extra pure (Scharlau, Barcelona, Spain), iron chloride (III) (VWR, Leuven, Belgium), disodium hydrogen phosphate dihydrate, sodium dihydrogen phosphate monohydrate, L(+)-ascorbic acid, trichloroacetic acid (Scharlau, Barcelona, Spain), ethanol 96% (Chimreactiv SRL, Bucharest, Romania), Trolox 97% (Acros Organics, Thermo Fisher Scientific, Pittsburgh, PA, USA), 2,2-Diphenyl-1-picrylhydrazyl (DPPH), gallic acid, Folin Ciocalteu's phenol reagent (Sigma Aldrich, Merck Group, Darmstadt, Germany), sodium bicarbonate (Reactivul București, Bucharest, Romania) and sodium chloride (Labkem, Barcelona, Spain). For β-glucan deter-

mination, the β-glucan Assay Kit (Yeast and Mushroom, Megazyme, Bray, County Wicklow, Ireland) was used.

*2.2. Methods*

2.2.1. Sample Preparation and Processing

The samples were prepared by mixing different volumes of homogenized slurry SBY in distilled water to obtain samples of different concentrations of SBY, as summarized in Table 1. The dry substance was determined gravimetrically, after freeze-drying the yeast slurry biomass in a ScanVac Cool Safe 55-4 lyophilizer (LaboGene, Allerod, Denmark). The experimental results are expressed per mass of slurry biomass dry weight (dw). The SBY content was varied according to Table 1.

**Table 1.** Preparation of SBY samples.

| Sample | SBY 'As Is' Concentration (% *v/v*) | SBY 'As Is' Concentration (mg/mL) | SBY 'Dry Basis' Concentration (mg/mL) |
|---|---|---|---|
| C1 | 6.25 | 63.75 | 9.85 |
| C2 | 12.50 | 127.50 | 19.70 |
| C3 | 25.00 | 255.00 | 39.67 |
| C4 | 37.5 | 382.50 | 59.30 |

Each of the prepared samples was subjected to ultrasonication using an ultrasonic liquid processor (Sonics VCX-750 Vibracell Ultrasonic Liquid Processor, Sonics & Materials, Inc. Newtown, CT, USA). The process was carried out for 30 min at 750 W power and 20 kHz frequency (ultrasound intensity of 9.375 W/mL or 16.875 J/mL), 40 °C and duty cycle (DC) of 33.33%, with 80% amplitude of the sonotrode.

The optimal SBY concentration was selected for testing the effect of different temperatures and DC, as shown in Table 2.

**Table 2.** Variation of ultrasonication temperature and duty cycle in SBY lysis.

| Sample | Temperature (°C) | Duty Cycle (%) |
|---|---|---|
| a40 | 40° | 33.33% |
| a50 | 50° | 33.33% |
| a60 | 60° | 33.33% |
| a70 | 70° | 33.33% |
| b40 | 40° | 50% |
| b50 | 50° | 50% |
| b60 | 60° | 50% |
| b70 | 70° | 50% |

2.2.2. Preparation of Spent Brewer's Yeast Extracts and Cell Walls

After lysis, the processed samples were centrifuged for 10 min at $7350 \times g$ to obtain the spent brewer's yeast extracts (SBYEs, supernatant) and cell walls (SBYCWs, pellet) from each sample. The SBYCWs were freeze-dried at −52 °C using a ScanVac Cool Safe 55-4 lyophilizer and resuspended in water to the desired concentration prior to each analysis.

2.2.3. Total Protein Assay of Spent Brewer's Yeast Extracts

The protein content in SBYEs was determined by Biuret assay against bovine serum albumin (BSA) standard curve, prepared within a concentration range of 0–8 mg/mL. Biuret reagent was added to the standard dilutions or samples at a ratio of 5:1. After 30 min of reaction at room temperature, absorbance was measured at λ = 550 nm using a microplate spectroscopy reader (CLARIOstar, BMG LABTECH, Ortenberg, Germany). The results were expressed as mg BSA equivalent/g dw SBY.

2.2.4. Disruption Efficiency Determination

Disruption efficiency (DE) was determined using the formula described by [24] (Equation (1)) based on the loss of dried biomass during ultrasonication:

$$DE = \frac{m_0 - m_1}{m_0} \tag{1}$$

where $m_0$ = dried biomass of sample before ultrasonic lysis and $m_1$ = residual dried biomass after ultrasonic lysis and centrifugation. A quantity of raw dried biomass equal to that resuspended in the case of each sample was freeze-dried using a ScanVac Cool Safe 55-4 lyophilizer (LaboGene, Allerod, Denmark) and weighed after drying to obtain the m0 values. The pellets of the centrifuged ultrasonicated samples were dried in a similar manner to obtain the m1 values.

2.2.5. FT-IR Assay of Spent Brewer's Yeast Cell Walls

The IR spectra were recorded with an IRTracer-100 spectrophotometer (Shimadzu, Japan) equipped with an Attenuated Total Reflectance (ATR) accessory. Sample recordings were performed at a wavelength range between 4000 and 400 cm$^{-1}$ with a resolution of 4 cm$^{-1}$.

2.2.6. Determination of the β-glucan Content in Spent Brewer's Yeast Cell Walls

The β-glucan content in SBYCWs was determined using the β-glucan Assay Kit (Yeast and Mushroom, Megazyme). The assay was performed on the yeast cell walls obtained from the optimal parameters, which were considered to have undergone the most efficient lysis. The percent of β-glucan was determined as the difference between the percent of total glucan and α-glucan, using the calculation formulas described in the manufacturer's manual.

2.2.7. Optical Microscopy of SBY and SBYCWs

To remove the cell debris, the SBYCWs subjected to optical microscopy were pre-washed according to the protocol described by [19,25], with some modifications. The pellet was washed three times with distilled water, three times with NaCl (17 mM, 34 mM and 85 mM) and once with ultrapure water.

Optical microscopies of yeast cells prior to and after lysis, as well as of SBYCW samples were performed using a Leica DM1000 LED optical microscope (Leica Microsystems, Wetzlar, Germany) equipped with an ICC50W digital camera, using a 40× objective. Methylene blue (1%) was used for yeast cell staining.

2.2.8. Scanning Electron Microscopy of SBY and SBYCWs

To remove the cell debris, the SBYCWs subjected to SEM analysis were pre-washed as described for optical microscopy in Section 2.2.7.

TM4000Plus II tabletop electron microscope (Hitachi, Tokyo, Japan) was used to visualize the spent brewer's yeast cell walls. The images were obtained at a 15 kV voltage, on a 100 μm scalebar, at a working distance of approximately 6 mm. A backscattered electron (BSE) detector along with standard (M) vacuum mode was used according to the producer's instructions.

2.2.9. Antioxidant Activity of Spent Brewer's Yeast Extracts and Cell Walls
Free Radical Scavenging Activity Assay Using the DPPH Method

The AOAs of SBYEs and SBYCWs were assessed using the DPPH (2,2-diphenyl-1-picrylhydrazyl) method described by [26]. The determination was performed against a Trolox standard curve within a 0.0125–0.15 mM Trolox range. The DPPH reagent, 0.3 mM in 96% ethanol (*v/v*), was added to the standard dilutions and samples in a 1:1 ratio. The reaction was allowed to occur for 30 min. The samples were centrifuged at 2900× *g* for 5 min in a Microspin 12 mini-centrifuge (Biosan, Riga, Latvia). The absorbance was

spectrophotometrically measured at λ = 517 nm. The results were expressed as Trolox equivalent μM/g dw SBY or SBYCW.

Potassium Ferricyanide Reducing Power (PFRAP) Assay

The PFRAP method described in [27,28] was adapted to evaluate the AOAs of SBYEs and SBYCWs. A standard curve of ascorbic acid was prepared in the range of 20–80 μg/mL. The standard dilutions and samples were mixed with 2.2 M phosphate buffer (pH = 6.6) and 1% potassium ferricyanide (1:2:2 ratio). The reaction mixtures were incubated for 20 min at 50 °C. Then, 10% TCA (trichloroacetic acid) was added to the reaction mixture in a ratio of 2:5. Following centrifugation (10 min, $6500\times g$, Microspin 12), the supernatant was mixed with distilled water in a 1:1 ratio. A 0.1% quantity of Iron chloride (III) was added to this mixture at a ratio of 1:10. The absorbance was measured at λ = 700 nm. The results were expressed as ascorbic acid equivalent μg/dw g SBY or SBYCW, respectively.

### 2.2.10. Total Polyphenol Content Assay

The total polyphenol content assay was adapted after the method described in [26] against a gallic acid standard curve. The gallic acid standards were prepared from a stock solution of 1 mg/mL in the range of 2–10 μg/mL. The Folin-Ciocalteu reagent was added to samples and standards mixed with water (1:9:1). The microplate in which the experiment was performed was stirred for 5 min using a microplate reader equipped with shaking functions (double orbital shaking, 400 rpm). $Na_2CO_3$ (7%) and water were added to the mixture and the reaction was allowed to occur for 60 min at room temperature. Spectrophotometric measurements were performed at λ = 765 nm. The total polyphenol content in the samples was determined as gallic acid equivalent (GAE) μg/mg SBYCW.

### 2.2.11. Statistical Analysis

IBM SPSS Statistics for Windows, version 26 (IBM SPSS Corp, Armonk, NY, USA), was used for statistical analysis of the results. To observe the statistical differences between samples, a one-way analysis of variance (ANOVA) test was used. Tukey honestly significant difference (HSD) was used to describe the significant statistical differences between samples. Pearson correlations between Biuret, DE, PFRAP of SBYEs, and DPPH of SBYEs were performed using SPSS Bivariate Correlations, while Matrix Scatter Plot was obtained using SPSS Chart Builder. A Heatmap was generated by Conditional Formatting using Microsoft Excel.

### 3. Results

*3.1. Streamlining Ultrasonication for Spent Brewer's Yeast Cell Lysis*

3.1.1. Influence of Spent Brewer's Yeast Concentration, Ultrasonic Temperature and Duty Cycle on Ultrasonic Lysis

Quantifying the intracellular compound release (e.g., proteins) is a method of evaluating lysis efficiency. The higher the intracellular compound release, the higher the lysis efficiency [29]. Quantification of the total protein content in SBYEs was performed using the Biuret assay. The results (Figure 1A) highlight a trend of descending protein content in SBYE with the increase in the SBY concentration, which indicates that ultrasonic lysis was more efficient in samples with lower cell density.

In terms of released protein content, the most efficient lysis occurred when ultrasonication was performed on the most diluted sample tested: C1—6.25% (*v*/*v*) SBY (40 °C, 33.33% DC). The extract obtained from this sample had the highest protein content, according to the Biuret assay results—342.565 mg proteins/g dw of SBY. The SBYE with the lowest protein content—131.433 mg proteins/g dw SBY—was obtained from the second most concentrated sample of SBY, 25% (*v*/*v*). Statistical analysis showed significant differences between the optimal C1 sample and the more concentrated samples.

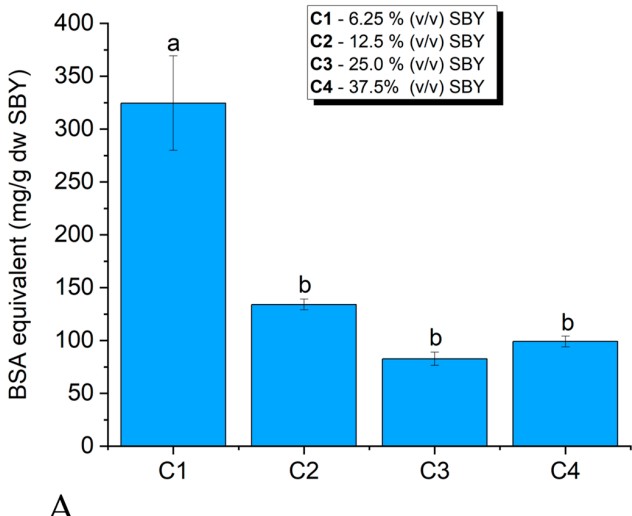
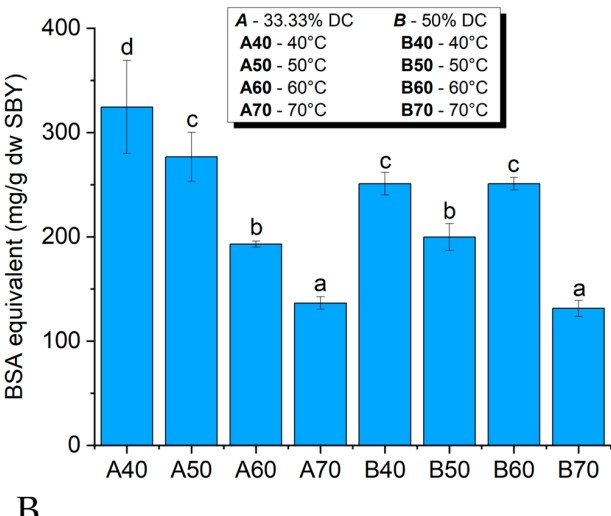

**Figure 1.** The total protein content in spent brewer's yeast (SBY) extracts (expressed as mg of bovine serum albumin (BSA) equivalent per dry weight (dw) of SBY) from ultrasonicated samples of (**A**) different SBY concentrations and (**B**) different temperatures and duty cycles. Sample a40 is the same as sample C1 from Figure 1A; different letters (a, b, c, d) represent statistically significant differences between samples at $p < 0.05$.

As the optimum concentration of those tested for the ultrasonic lysis of the SBY cells was considered to be 6.25% (*v/v*) SBY, further analysis was conducted using samples at this constant concentration to observe the influence of duty cycle and temperature on the lysis efficiency (Figure 1B). For this purpose, the Biuret assay was performed on SBYEs obtained after the ultrasonication of the samples at different temperatures (40 °C, 50 °C, 60 °C, and 70 °C) and duty cycles (33.33% and 50%).

At the lowest temperatures tested—40 °C and 50 °C—the amount of released intracellular proteins detected increased when ultrasonication was performed using a lower DC of 33.33% compared to a DC of 50%. At 60 °C, higher protein content was obtained at 50% DC than at 33.33% DC and 70 °C, the protein contents at 33.33% and 50% DC were similar. The temperature significantly influenced the lysis process at 33.33% DC. The extracts had higher protein content at the lowest processing temperatures, i.e., 40 °C and 50 °C. The lowest protein content in SBYEs was observed at the highest temperature tested, 70 °C. The negative impact of high temperature with respect to the protein content in SBYEs was statistically significant. The trend at 50% DC was somewhat different from that at 33.33%. Relatively high protein contents were obtained up to 60 °C and became significantly diminished only at 70 °C. The optimum lysis conditions were 33.33% DC and 40 °C (sample C1/a40).

### 3.1.2. Disruption Efficiency

As high temperatures could cause protein denaturation and precipitation and affect its soluble content, evaluating lysis efficiency through quantifying intracellular protein release should be confirmed using an additional method. In this work, a gravimetric method was used to verify the previous results based on protein content.

The disruption efficiency was calculated using Equation (1), based on the dry biomass loss after ultrasonication. The results (Figure 2) confirmed the previous results, which suggests that the lysis efficiency is almost inversely proportional to sample concentration and ultrasonication temperature.

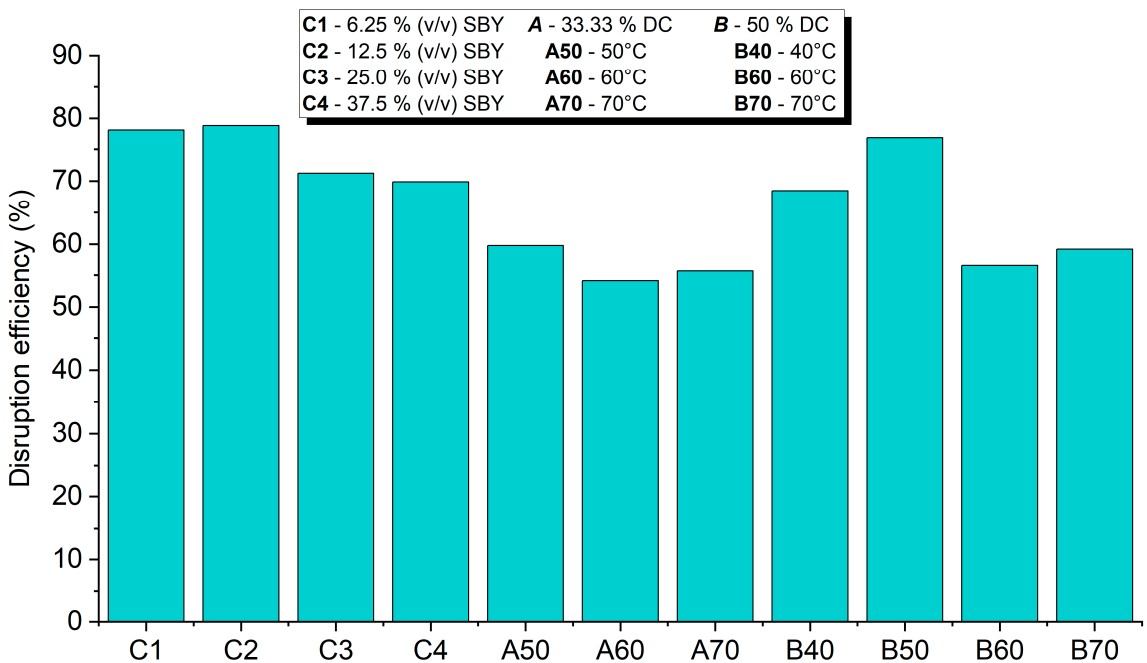

**Figure 2.** Ultrasonication lysis efficiency of spent brewer's yeast samples based on dry biomass loss during ultrasonic lysis.

### 3.2. FT-IR Assay of Disrupted Spent Brewer's Yeast Cells and Yeast Cell Walls

The raw material (SBY), the optimal sample (C1), and the cell walls (SBYCWs) from C1 were subjected to FT-IR analysis (Figure 3).

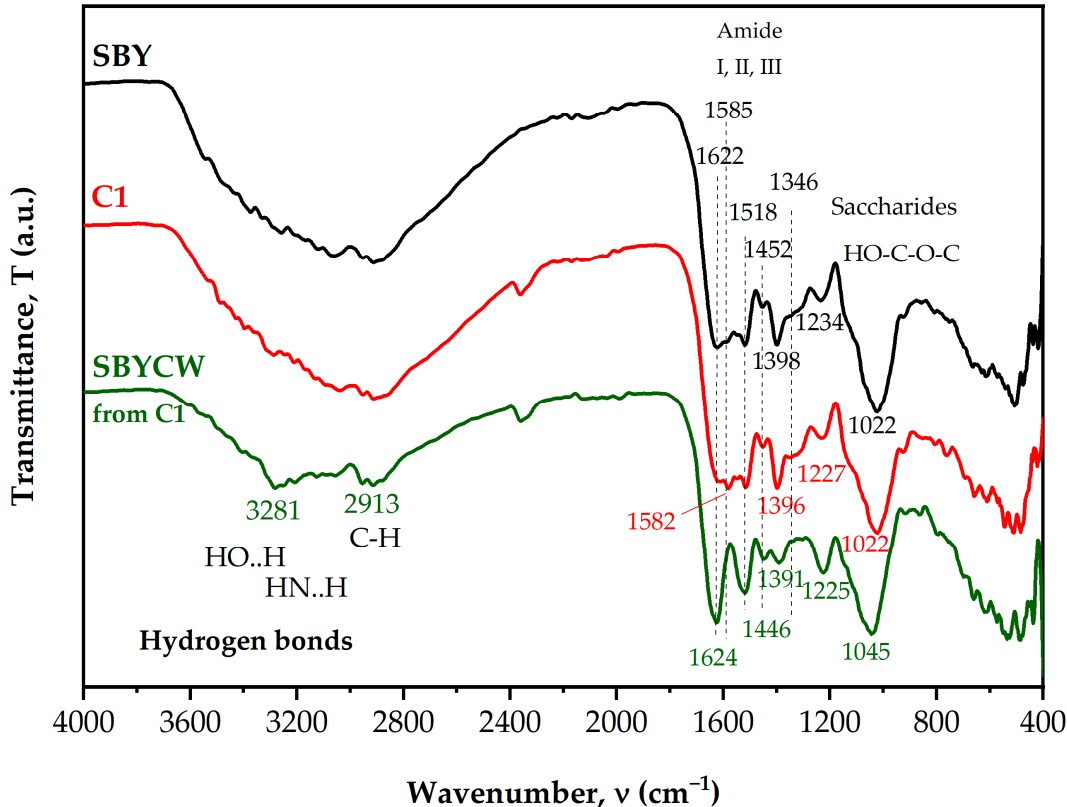

**Figure 3.** Intercalated FT-IR spectrum for raw spent brewer's yeast (SBY), ultrasonicated spent brewer's yeast (C1) and yeast cell walls (SBYCWs).

FT-IR assay of yeasts presented peaks corresponding to proteins, polysaccharides, lipids, and nucleic acids, which are the main components of this biomass [30,31]. The SBY sample presented the characteristic bands for yeasts: the large band in the region 3200–3500 $cm^{-1}$ corresponded to -OH and -NH groups involved in H-bonds [32], 3000–2800 $cm^{-1}$ corresponded to the -$CH_2$ and -$CH_3$ groups from aliphatic chains such as lipids [32–34], 1622 and 1518 $cm^{-1}$ corresponded to amide I and amide II in proteins and chitin, respectively, and 1022 $cm^{-1}$ corresponded to polysaccharides, mainly β-glucans [32–36]. Other bands are 1585 $cm^{-1}$ and 1346 $cm^{-1}$, which most probably come from the N-acetylglucosamine residues in mannoproteins and chitin [34,36–38]; 1452 $cm^{-1}$, which was previously assigned to -$CH_2$ and -$CH_3$ in aliphatic compounds; side chains of amino acids; 1398 $cm^{-1}$, previously assigned to carboxylic groups and $CH_2$ wagging in lipids and β-(1,3)-glucans [32]; and 1234 $cm^{-1}$, suggested to reflect phosphate groups present in nucleic acids, phospholipids and/or proteins [35,39,40]. Amide III also has a contribution in the region 1350–1250 $cm^{-1}$. Upon lysis, there were some changes in the spectral bands, the most significant being the following band shifts: 1622 to 1618 $cm^{-1}$, 1585 to 1582 $cm^{-1}$, 1398 to 1396 $cm^{-1}$, and 1234 to 1227 $cm^{-1}$. Additionally, the bands at 1582/1585 $cm^{-1}$ and 1346 $cm^{-1}$ were more intense than in the sample analyzed before lysis. Some bands remained unchanged: 1518, 1452, 1346 and 1022 $cm^{-1}$. More significant differences were observed when comparing the SBY sample with the SBYCW sample: bands 1585 $cm^{-1}$ and 1346 $cm^{-1}$ disappeared, bands 1622, 1518 and 1234 $cm^{-1}$ became more intense and well defined and band 1398 $cm^{-1}$ became slightly broader. There were also some band shifts in SBYCWs compared to SBY: 1622 to 1624 $cm^{-1}$, 1452 to 1446 $cm^{-1}$, 1398 to 1391 $cm^{-1}$, 1234 to 1225 $cm^{-1}$ and 1022 to 1045 $cm^{-1}$.

### 3.3. β-glucan Content in Spent Brewer's Yeast Cell Walls

The β-glucan content was determined using the SBYCWs from the optimal sample with regard to disruption efficiency (sample C1/a40—6.25% SBY, 40 °C, 33.33% duty cycle) using the β-glucan Assay Kit (Yeast and Mushroom, Megazyme) against the control provided by the kit, with a 49.9% β-glucan stated by the manufacturers. The β-glucan content of the optimal sample was 41.3%, which shows a relatively high purity of the SBYCWs resulting from an efficient lysis process.

### 3.4. Optical Microscopy of SPY and SPYCWs

Optical microscopy of lysed yeast cells and SBYCWs confirmed the occurrence of cell lysis. Methylene blue staining allowed visualization of intact yeast cells in SBY before lysis (Figure 4A,B) and the released intracellular content after lysis (Figure 4C,D). SBYCWs appeared as white, almost round circles isolated from the rest of the cell debris, with some being clustered in aggregates (Figure 4C–F). Part of the cells were still intact after lysis.

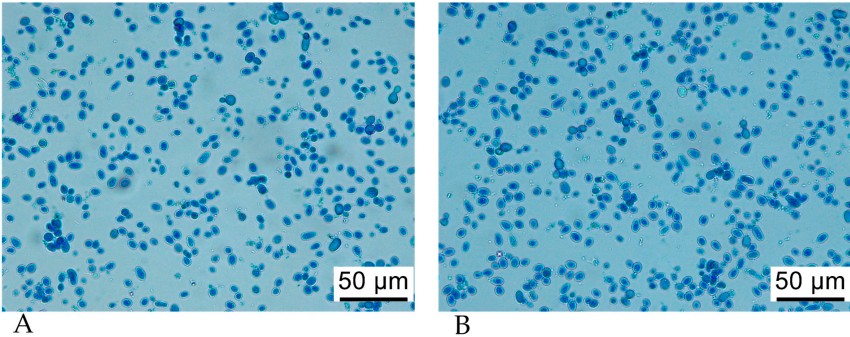

**Figure 4.** *Cont.*

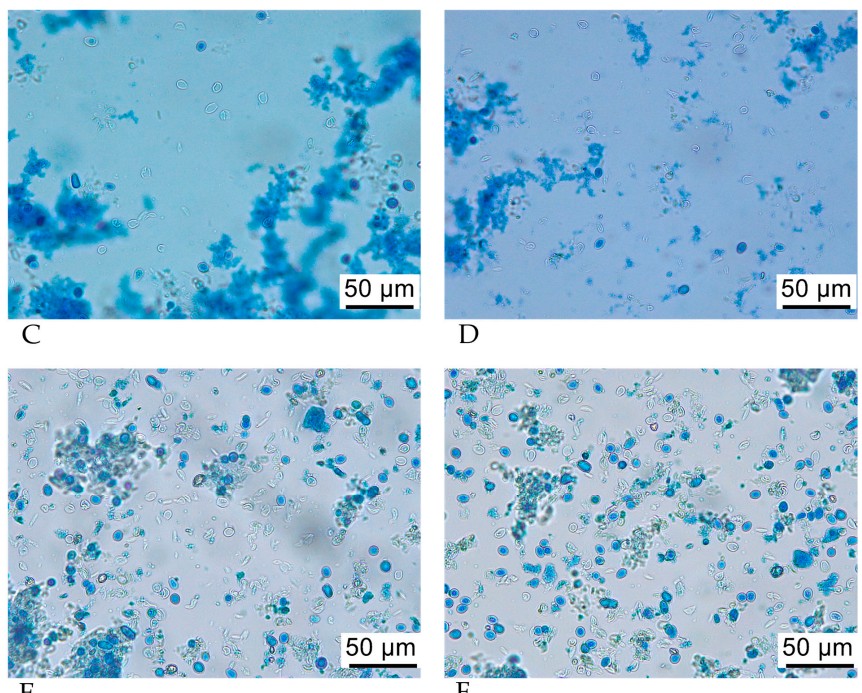

**Figure 4.** Optical microscopy images of the C1 sample (6.25% SBY) prior to lysis (**A**,**B**) and after lysis (**C**,**D**) and of SBYCWs after washing with NaCl (**E**,**F**). Staining of yeast cells was performed using 1% methylene blue.

### 3.5. Scanning Electron Microscopy of SPY and SPYCWs

The SEM micrographs present images of the sample yeast cells before lysis (Figure 5A) and after lysis (Figure 5B) and yeast cell walls obtained from ultrasonic lysis (Figure 5C,D).

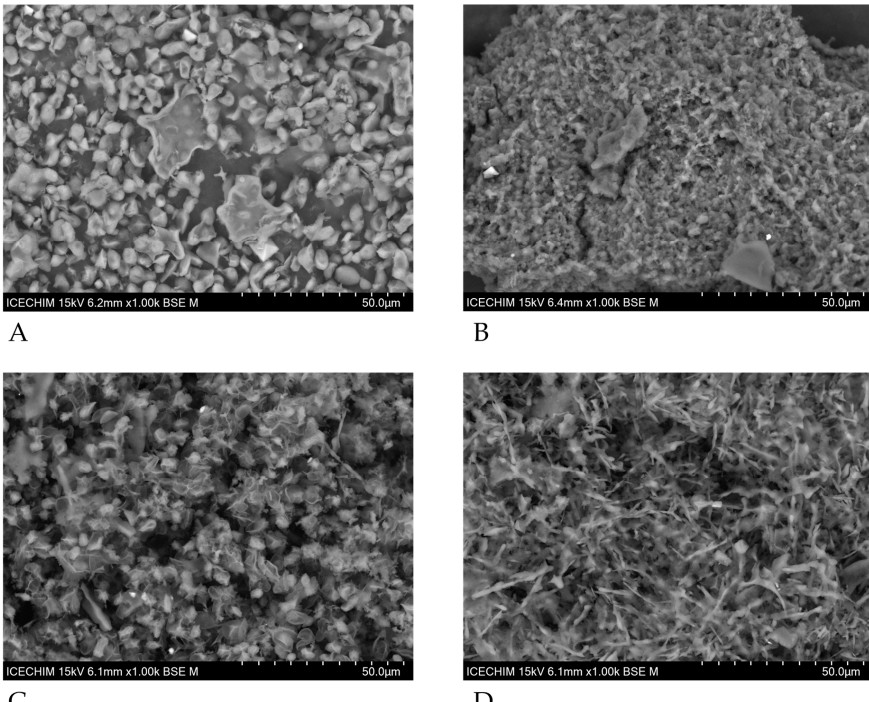

**Figure 5.** SEM micrographs of spent brewer's yeast cells: sample C1 prior to ultrasonication (**A**) and after sonication (**B**) and the spent brewer's yeast cell walls collected after ultrasonication and centrifugation (**C**,**D**).

The unprocessed SBY cells are intact and morphologically normal. SEM confirms that the lysis of yeast cells was efficient, as the disrupted cell membranes are visible in SEM micrographs of yeast cell walls, with obvious breakages and changes in shape. Together with optical microscopy, SEM provides images of the structural changes that occur at a cellular level during ultrasonic disruption.

*3.6. Antioxidant Activity of Spent Brewer's Yeast Extracts and Cell Walls*

3.6.1. Antioxidant Activity of Spent Brewer's Yeast Extracts

The results of the AOA of SBYEs using PFRAP and DPPH methods are presented in Figure 6. Both assays showed significant antioxidant activity of SBYEs. The strongest free radical scavenging capacities were observed in SBYEs from samples that were subjected to the highest efficient lysis processes. Both DPPH and PFRAP assays showed significant differences between SBYEs collected from the 6.25% (*v/v*) SBY sample in comparison to those collected from highly concentrated samples.

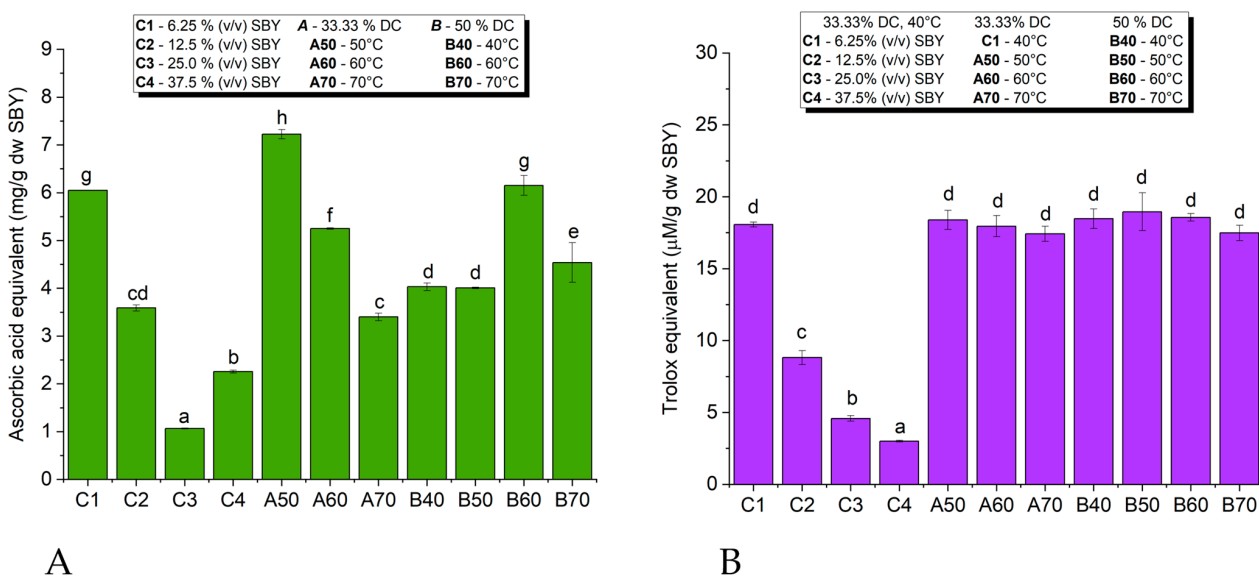

**Figure 6.** Antioxidant activity of spent brewer's yeast extracts based on PFRAP (**A**) and DPPH assays (**B**); different letters (a, b, . . ., h) represent statistically significant differences between samples at *p* < 0.05.

However, with respect to the DC and temperature, the PFRAP results showed a higher correlation with lysis efficiency than the DPPH results. The differences between the AOAs of the samples processed at different temperatures and DCs were more significant in the PFRAP assay compared to the DPPH assay. At 33.3% DC, the maximum PFRAP AOA was recorded for the A50 sample (lysis at 50 °C), with higher temperatures inducing a significant decrease in PFRAP AOA. The behavior was opposite at 50% DC, with higher lysis temperatures inducing higher PFRAP AOA. At 40 °C and 50 °C, the PFRAP AOAs were significantly higher at 33.33% DC compared to 50% DC. At 60 °C and 70 °C, the PFRAP AOAs were higher at 50% DC compared to 33.33% DC. Neither the temperature nor the DC influenced the DPPH AOA of SBYEs.

Pearson correlation was used to determine the relationships of direct proportionality or inverse proportionality from a statistical point of view. The matrix scatterplot gives us an overview of the correlation between the different analyses. There were certain patterns that had an ordered structure, such as PFRAP–Biuret, DPPH–Biuret and DPPH–PFRAP, which indicated a statistically significant correlation (Figure 7).

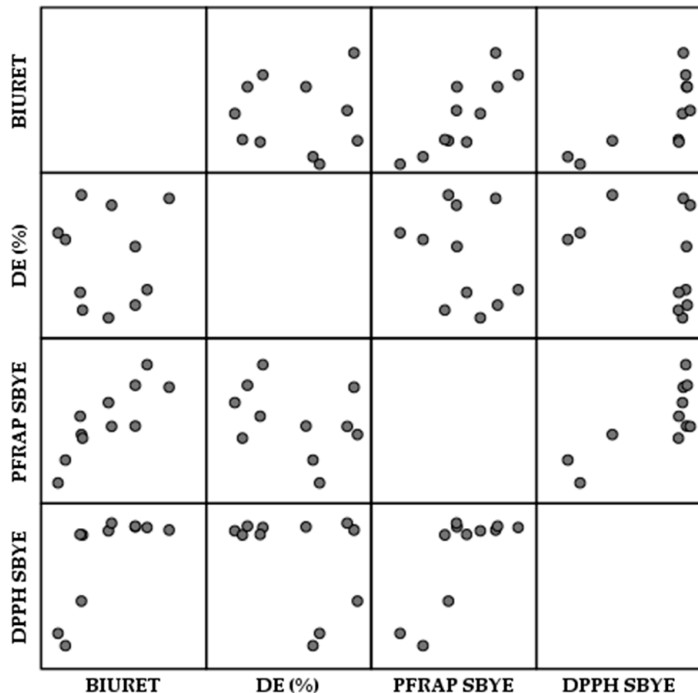

**Figure 7.** Correlation matrix scatterplot of the experimental variant analysis for SBYEs. The independent parameters yeast concentration, temperature and duty cycle (DC) were considered.

Pearson correlation was combined with the Heatmap in order to provide a more detailed representation. Table 3 indicates that there was no statistically significant correlation between DE (%) and the other analyses, although there was a slight negative correlation between DE and both PFRAP and DPPH.

**Table 3.** Pearson correlation coefficients of the experimental variant analysis for SBYEs. The independent parameters yeast concentration, temperature and duty cycle (DC) were considered.

| | BIURET | DE (%) | PFRAP SBYE | DPPH SBYE | | −1 |
|---|---|---|---|---|---|---|
| BIURET | 1 | | | | | |
| DE (%) | 0.01 | 1 | | | | 0 |
| PFRAP | 0.842 ** | −0.328 | 1 | | | |
| DPPH | 0.718 * | −0.385 | 0.765 ** | 1 | | 1 |

** Correlation is significant at the 0.01 level (2-tailed). * Correlation is significant at the 0.05 level (2-tailed).

There was a direct proportional relationship between protein concentration (Biuret) and antioxidant activity, i.e., the correlation between Biuret and PFRAP was significant at the 0.01 level, while the correlation between Biuret and DPPH was significant at the 0.05 level. There was also a statistically significant direct proportional relationship at the 0.01 level between the two antioxidant activities, DPPH and PFRAP.

3.6.2. Antioxidant Activity of Spent Brewer's Yeast Cell Walls

As the diluted SBY samples were considered to be the most efficiently lysed, the pellets resulting from the centrifugation of the processed samples were considered to have the highest content of yeast cell walls based on the intracellular protein release and disruption efficiency confirmed by previously presented results.

All SBYCW samples showed AOAs based on the PFRAP and DPPH assays, as shown in Figure 8.

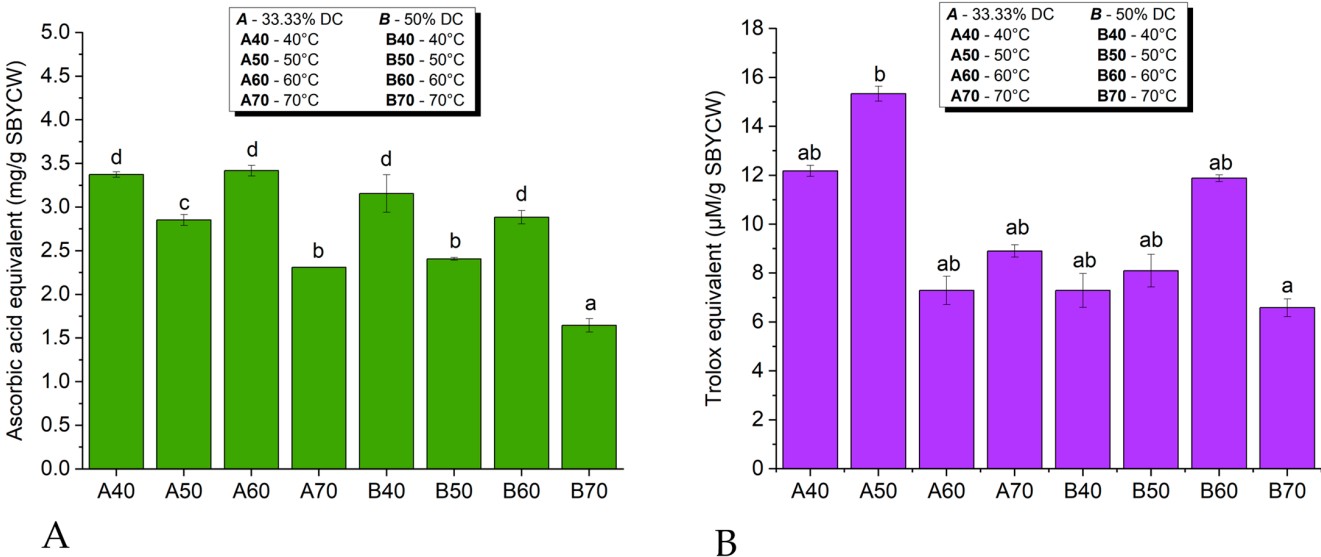

A

B

**Figure 8.** Antioxidant activity of spent brewer's yeast cells walls based on PFRAP (**A**) and DPPH (**B**) assays; different letters (a, b, c, d) represent statistically significant differences between samples at $p < 0.05$.

AOA depended on all parameters, the AOA method, the temperature and the DC of lysis. The dependence of PFRAP on temperature was similar at 33.33% DC and 50% DC. The PFRAP AOA of SBYCWs obtained with 33.33% DC was generally higher than that of SBYCWs obtained with 50% DC. In the case of DPPH, the highest AOA was obtained for the A50 sample, which was significantly higher than for all the other samples, irrespective of DC. When using 50% DC, the maximum DPPH was obtained for sample B60, although this was lower than for sample A50.

The results of the total polyphenol content (TPC) assay (Figure 9) showed that SPBY-CWs retain some polyphenols in various amounts, depending on the lysis parameters.

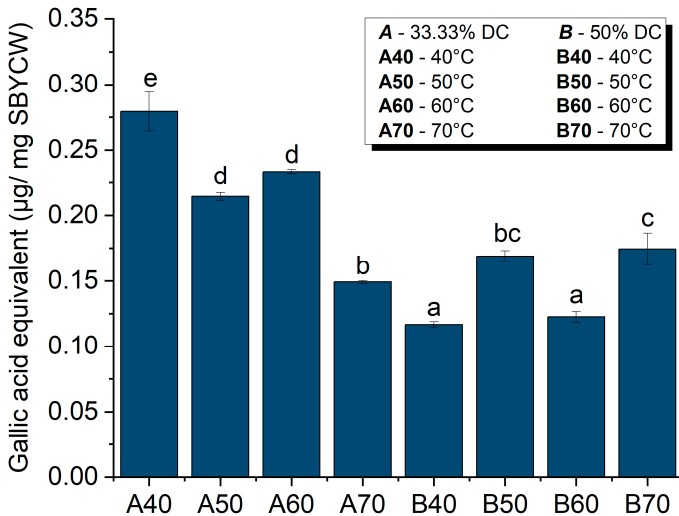

**Figure 9.** Total polyphenol content of spent brewer's yeast cell walls. Values of the columns with different letters (a, b, c, d, e) are statistically significant differences between samples at $p < 0.05$.

We analyzed Pearson correlation again, this time taking into consideration only the varied parameters common to all responses, i.e., temperature and DC, in order to include the correlation analysis of the cell walls (Figure 10 and Table 2).

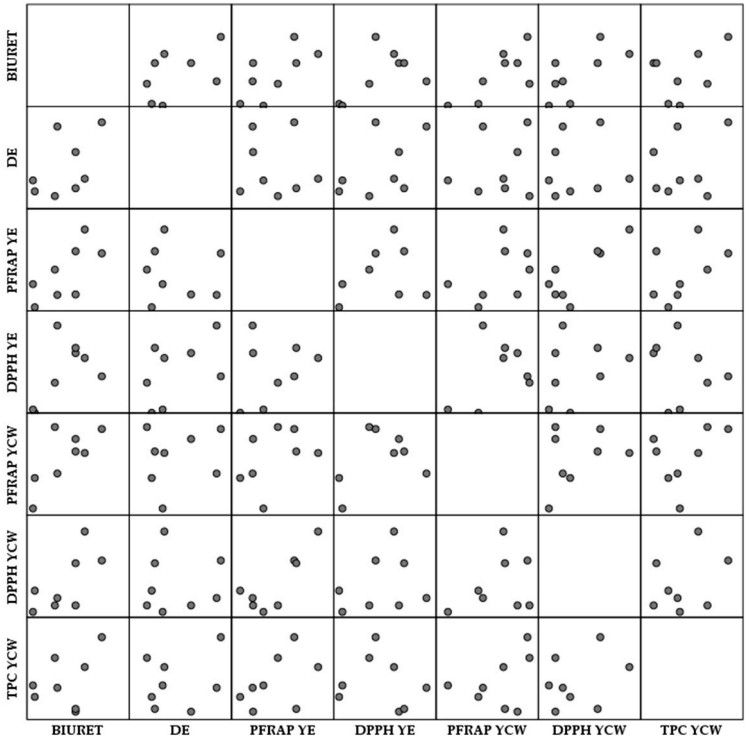

**Figure 10.** Correlation matrix scatterplot of experimental variant analyses for SBYEs. The independent parameters yeast concentration, temperature and duty cycle (DC) were considered.

In this case, statistically significant positive correlations were obtained only between PFRAP SBYCWs and Biuret SBYEs and between DPPH SBYCWs and PFRAP SBYEs. Most of the other correlation coefficients were positive, except for PFRAP SBYE–DE (%) and TPC SBYCW–DPPH SBYE, which were slightly below zero. There was no statistically significant negative correlation in Table 4.

**Table 4.** Pearson correlation coefficients of experimental variant analysis for SBYEs and SBYCWs. The independent parameters yeast concentration, temperature and duty cycle (DC) were considered.

| | BIURET | DE (%) | PFRAP SBYE | DPPH SBYE | PFRAP SBYCW | DPPH SBYCW | TPC SBYCW | | −1 |
|---|---|---|---|---|---|---|---|---|---|
| BIURET | 1 | | | | | | | | |
| DE (%) | 0.502 | 1 | | | | | | | |
| PFRAP SBYE | 0.69 | −0.075 | 1 | | | | | | |
| DPPH SBYE | 0.538 | 0.494 | 0.247 | 1 | | | | | 0 |
| PFRAP SBYCW | 0.738 * | 0.178 | 0.425 | 0.356 | 1 | | | | |
| DPPH SBYCW | 0.69 | 0.045 | 0.832 * | 0.266 | 0.318 | 1 | | | |
| TPC SBYCW | 0.391 | 0.268 | 0.484 | −0.179 | 0.386 | 0.324 | 1 | | 1 |

* Correlation is significant at the 0.05 level (2-tailed).

## 4. Discussion

Ultrasonication is a highly efficient lysis method. Nevertheless, to obtain elevated degrees of cell disruption, it is important to determine and adapt the processing parameters to the type of sample and microorganism [41,42].

Studies on the influence of ultrasonication parameters on SBY cell lysis efficiency and protein release are scarce. However, the literature describes temperature and duty cycle as

among the most impactful parameters in yeast cell ultrasonication [29,43]. According to [26], the release of proteins following ultrasonication is not influenced by sample concentration, but rather by acoustic power input. Other studies point out that the cell suspension concentration is a significant parameter in ultrasonic lysis [27]. Overall, the optimum lysis conditions depend directly on the energy that is delivered to the processed cell suspension. When the cell density increases, the energy transmitted to each cell consequently decreases. Still, it is important to take into consideration that lower cell densities might lead to fewer chances of collision among the cells. For that reason, it is important to determine how the concentration of a particular type of biomass might be correlated to the disruption efficiency [44,45]. In the current study, the sample concentration had a notable influence on the disruption efficiency in terms of protein release from lysed SBY cells, the lower the tested concentration, the higher the protein release. Most of the reported results on the influence of ultrasonication temperature state that an increase in this parameter will negatively affect the disruption efficiency, especially in terms of biomolecule release, such as proteins and polysaccharides [46]. Our study strongly confirms this data, as the protein content of the obtained yeast extracts was significantly lower when the processing temperature was increased (e.g., 60 °C or 70 °C). The extracts obtained from SBY after ultrasonication generally had elevated protein content in comparison to values reported in the literature. Jacob et al. reported that yeast extract obtained through bead milling, ultrasonication and autolysis had a protein content of 321.56 mg/g, 285.40 mg/g and 102.00 mg/g [18,47]. The protein content of the SBYEs we obtained after ultrasonication ranged between 131.433 and 324.565 mg/g dw SBY, which confirmed once again that SBYEs obtained through ultrasonication are a great source of proteins. Moreover, the optimal ultrasonication conditions resulted in the highest protein content, which is comparable with the elevated values reported in the literature, generally obtained through autolysis or enzymatic hydrolysis. Podpora et al. reported a protein content of 365 mg/g in yeast extracts derived from enzymatic lysis [48]. Another study by Podpora et al. reports a value of 303 mg/g proteins in yeast extracts obtained through autolysis [49]. However, these lysis methods require much longer processing times and/or the use of expensive enzymes, which are not needed in the ultrasonic process that we described. It is important to assess the ultrasonication process taking into consideration the thermolabile nature of proteins and other released biocompounds with nutritive value [50]. As the goal of this ultrasonication process was to obtain spent yeast derivatives with bioactive value, it is, therefore, necessary to use processing temperatures that contribute to the occurrence of yeast cell disruption while avoiding the degradation of the valuable released biocompounds.

Longer duty cycles are frequently associated with more efficient disruption degrees. Still, although some studies suggest that an increase in DC is proportional to intracellular protein release during ultrasonication, they also show that a decrease in DC might prevent sample overheating and protein degradation. This might explain the enhanced protein content of some of the SBYEs from samples lysed at lower DCs [29]. Since the aim of this study was to optimize the ultrasonication parameters for obtaining yeast derivatives with elevated nutraceutical values, a higher content of bioactive compounds, lower concentration, duty cycle and temperature were considered to be optimal parameters, taking into consideration the protein content and antioxidant activities of the yeast extracts.

Besides quantifying the released biocompounds and biomass loss, imaging approaches such as optical or electron microscopy are usually used to assess yeast cell disruption [24,25]. Very few studies reported images of lysed spent yeast cells. Øvrum Hansen et al. [51] presented SEM micrographs of disrupted spent baker's yeast cells, which provided clear images of the breakage of autolyzed yeast cells. To the best of our knowledge, there are no SEM micrographs showing the changes suffered by SBY cells after ultrasonic disruption. The current study provides a clear image of ultrasonic-disrupted SBY cell walls in comparison to intact SBY cells prior to lysis. The SBYCWs obtained in this study show hollow, partially aggregated spherical shapes as well as cell wall fragments similar to other lysed yeast cells. These are also visible in the SEM micrographs.

The yeast cell wall is a complex structure of polysaccharides and proteins, specifically β-glucan, mannoproteins and chitin. Yeast β-glucan is known for its antioxidant, prebiotic and immunomodulatory properties [52–54] The β-glucan content in SBYCWs was assessed not only as an indicator of yeast cell wall content but also because it is a compound of major interest given its bioactive potential. The β-glucan content of yeast cell walls varies greatly, depending on the yeast source, accounting for roughly 30–60% of the dry weight [55]. We obtained 41.3% β-glucan content by assessing ice-dried yeast cell walls from the optimal C1 sample. Our results seem to correspond to the average values reported in the literature.

The changes in FT-IR spectra confirmed cell lysis and gave some insights into the composition and structural changes. Besides band shifts and intensity changes, one of the most important observations is related to the bands at 1585 cm$^{-1}$ and 1346 cm$^{-1}$, which become more distinct upon ultrasonication and disappear from the cell walls (SBYCWs). These bands correspond to N-acetylglucosamine or NH$_2$ from chitin/chitosan and mannoproteins and the changes suggest that the chitin and mannoproteins were partially removed from the cell walls or its interactions suffered significant modifications. The more clearly defined and intense bands in the region of amide I and II bands together with the bands corresponding to polysaccharides (1200–900 cm$^{-1}$) of SBYCWs suggest the presence of mainly polysaccharides and proteins (probably mannoproteins) in the cell walls upon ultrasonication.

Both DPPH and PFRAP assays showed elevated antioxidant activity of SBYEs, which is due to the presence of extracted bioactive compounds with AOA upon lysis. The AOA values of SBY derivatives reported in other studies are rather scarce and refer mainly to yeast extracts. Vieira et al. reported DPPH and PFRAP AOA mean values of $59.7 \pm 2.5$ and $127.6 \pm 1.0$ mg TE/100 g dw yeast [13]. The AOA values of our SBYEs, evaluated using similar assays, were significantly higher, with a maximum of 18.957 µM TE eq/g dw SBY based on the DPPH method and 6.05 mg Ascorbic acid eq/g dw SBY based on the PFRAP method from 6.25% SBY samples ultrasonicated at 50 °C and DC of 50% and 33.33%, respectively. The differences come from the processing method; the previous study separated the yeast cells from the beer liquor, while we used the slurry for the lysis process. The beer liquor probably contributes significant amounts of antioxidants to the slurry. Beer is known for its antioxidant activity, which is related mainly to polyphenols and melanoidins [56,57]. However, information about the presence of such antioxidants in the beer liquor associated with SBY is scarce in the literature and deserves further attention.

The antioxidant activity of yeast cell walls has been reported mostly in relation to their bioactive polysaccharides content, specifically mannoses and β-glucan, and thiol groups [58–62]. Several reports on wine-spent yeast and lees pointed to the presence of polyphenols, which are strong antioxidants (resulting mostly from biomass processing during the winery process), that seem to be adsorbed on the yeast cell wall surface [61]. Some studies describe a similar phenomenon in yeast resulting from brewery fermentation processes [13,40,41].

The significant decrease in AOAs of SBYEs and SBYCWs from samples processed at temperatures over 60 °C is a behavior also confirmed by [39], which mentioned the "high temperature-high antioxidant activity effect" as a consequence of optimum lysis conditions and explained that there is a certain temperature threshold beyond which the antioxidants are degraded.

Pearson correlation analysis gave some interesting results. When all independent parameters are taken into consideration, the AOA of the lysates (SBYEs) is relatively strongly and positively correlated with Biuret and DPPH is positively correlated with PFRAP of SBYEs at a statistically significant level. When the yeast concentration influence is not considered, this correlation is reduced, but it also becomes statistically non-significant. The difference indicates that yeast concentration has a similar influence on the extracted protein content and the two AOAs; this can be seen in Figures 1 and 6. As the AOA is given by various compounds, it indicates that their release is affected similarly by the yeast concentration. Other studies reported strong positive correlations between DPPH and PFRAP in sparkling wines [63] or weak positive correlations with hydrolyzed SBY

using various techniques [64], which suggests a heterogeneous and dynamic behavior of yeast-based systems that is highly dependent on various parameters.

Temperature and DC seem to not influence the AOA the same as the protein content, especially in the case of DPPH. This indicates that the various antioxidants extracted are impacted differently by these two independent variables—temperature and DC. This difference could be due to the impact on the extraction yield, compound stability or both. From Figures 1 and 6, one can observe that the highest protein contents are obtained at 40 !°C with 33.33% DC, 40 °C and 60 °C with 50% DC; however, the PFRAP AOA is highest at 50 °C with 33.33% DC and 60 °C with 50% DC. In the case of DPPH, the difference is even more evident, as the temperature and DC do not influence DPPH activity, which explains the lower correlation coefficient. We can conclude that in the case of DPPH, other compounds, which are not significantly affected by temperature and DC, give stronger activity compared to the proteins/peptides.

Another interesting aspect is the correlation between the AOA of the yeast cell walls obtained upon lysis and the lysate properties, which, to the best of our knowledge, we performed for the first time for these types of samples. Both AOA, i.e., PFRAP and DPPH, were each positively correlated with a specific property of SBYE, i.e., Biuret and PFRAP respectively. As previously mentioned, the insoluble fraction following lysis, represented by the cell walls, has AOA, mainly due to sulfur-containing compounds, mannoproteins, polysaccharides, and possibly, adsorbed polyphenols. The mechanisms underlying the two types of AOAs are different: DPPH involves the transfer of either or both H and electrons, while PFRAP only involves the transfer of electrons. In a previous study, it was shown that almost 90% of the DPPH activity of the *Saccharomyces cerevisiae* cell wall was attributed to the thiol groups, adsorbed polyphenols and mannans [61]. In the case of FRAP (which is similar to PFRAP), these compounds accounted for only around 54% of AOA, with the thiols contributing as low as 0.3% of the AOA. The authors concluded that other compounds contributed to the FRAP activity as well. Another characteristic mechanism, especially in the case of polysaccharides and (P)FRAP, is metal or radical chelation, including by chitin/chitosan [65,66]. In our case, we propose that cell lysis, especially the removal of proteins from the yeast cells, facilitates better contact between potassium ferricyanide and the polysaccharides and mannoproteins remaining in the cell wall, resulting in increased PFRAP activity. In the case of DPPH, the more compounds that give PFRAP reaction but less DPPH reaction are released, the more the SBYCWs are active towards DPPH. These compounds remain to be identified.

## 5. Conclusions

Spent brewer's yeast is revealed as a pool of easily obtainable bioactive compounds such as proteins, β-glucans and antioxidants. Yeast cells were efficiently disrupted through ultrasonic treatment. The process was streamlined by identifying optimal lysis conditions. The most efficient lysis, in terms of protein release and dry biomass loss, was achieved using the most diluted sample (6.25% (*v/v*) SBY) tested at 40 °C and 33.33% duty cycle. Optical microscopy, SEM and FTIR confirmed the lysis of SBY and gave morphological, structural and compositional information on yeast cell walls. The spent brewer's yeast derivatives resulting from disruption had elevated radical-scavenging and potassium ferricyanide reducing-power activities. Pearson correlation gave some new, intriguing results that should be investigated further. Overall, SBY processing via ultrasonication provides green alternatives for factory by-product handling, delivering derivatives of high nutraceutical value that could find great use in formulating products with health-promoting benefits.

**Author Contributions:** Conceptualization, L.T.C. and F.O.; methodology, L.T.C. and D.C.-A.; validation, D.C.-A., I.C.F. and F.O.; formal analysis, L.T.C. and D.C.-A.; investigation, L.T.C. and N.T.; resources, C.L. and R.N.N.; data curation, L.T.C. and C.L.; writing—original draft preparation, L.T.C.; writing—review and editing, D.C.-A. and F.O.; visualization, I.C.F. and F.O.; supervision, I.C.F. and F.O.; project administration, C.L. and F.O.; funding acquisition, R.N.N. and F.O. All authors have read and agreed to the published version of the manuscript.

**Funding:** This work was funded by project POC-A1-A1.2.3-G-2015-P_40_352-SECVENT, Sequential processes to close bioeconomy side stream and innovative bioproducts resulted from these, contract 81/2016, SMIS 105684, funded by cohesion funds of the European Union, subsidiary project 1743/2022-ToxiSorb.

**Institutional Review Board Statement:** Not applicable.

**Informed Consent Statement:** Not applicable.

**Data Availability Statement:** All the data are contained within the article.

**Acknowledgments:** We thank Luminița Dimitriu and Bogdan Trică for their assistance with statistical analysis, antioxidant activity and valuable discussions. The scanning electron microscope was acquired within the frame of the project 15PFE Next-Bexcel.

**Conflicts of Interest:** The authors declare no conflict of interest.

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
