# Peer review of "Valorization of Spent Brewer’s Yeast Bioactive Components via an Optimized Ultrasonication Process"

_fermentation, doi:10.3390/fermentation9110952_

Round 1

Reviewer 1 Report

Comments and Suggestions for Authors

Dear Authors,

the experiments are well done and the results indicating the best ultrasonic conditions for yeast’s cells lysis to extract functional compounds. However in my opinion the manuscript is observatory rather than explanatory. Many assays are done to confirm yeast cells disruption, but are redundant and no explanation of results is present. Furthermore, the results lack a comparison with literature data. I also suggest to summarize materials and methods and the statistical analysis presented in the results. 

Author Response

We appreciate your efforts to review our manuscript, and we are grateful for your valuable comments, which we hopefully managed to materialize in an improved version of our work.

Hence, we have revised the manuscript according to your comments and suggestions. The changes we have made are detailed below. Please note that when the position of the modification made within the text is specified, reference is made to the line number in the revised manuscript saved with the track changes option.

Comment 1: The experiments are well done and the results indicating the best ultrasonic conditions for yeast’s cells lysis to extract functional compounds. However, in my opinion the manuscript is observatory rather than explanatory. Many assays are done to confirm yeast cells disruption, but are redundant and no explanation of results is present. Furthermore, the results lack a comparison with literature data. I also suggest to summarize materials and methods and the statistical analysis presented in the results. 

Authors response to Comment 1:

The observatory perspective of the article might be present in some proportion since one of the goals of this work was to describe in-depth both the structural and chemical changes that happen during the ultrasonic disruption of the spent brewer’s yeast cells and the nutraceutical potential of spent yeast derivatives (antioxidant activity, protein content) in correlation with the ultrasonication conditions.

The many assays were performed not only to confirm spent yeast cells’ lysis but also to provide additional information about the phenomena that occur during the process and to outline a more comprehensive characterization in terms of the bioactive compound content of the resulted derivatives (proteins, β-glucan, antioxidants). The explanation of the obtained results is present mainly in the Discussions section. The variation of the protein content of the spent brewer’s yeast extracts is explained in relation to the varied parameters, specifically duty cycle and temperature (Lines: 416-421, 427-434). We have added more explanations of the results, in correlation with the literature data, especially in the discussion part (Lines: 421-427, 448-452), concerning the mechanism and justification of optimum lysis condition in spent brewer’s yeast cells.

The protein content was determined not only to confirm the occurrence of efficient cell lysis but also to point out the nutraceutical value of the resulted extract derivatives, given the high protein content, which is highlighted in the conclusion phrase (Line: 562-563). We added two more explanatory lines to highlight the idea (Lines: 438-440).

The microscopy assays were performed to provide a visualization of the structural changes which occur during ultrasonic disruption of yeast cells. These changes are detailed in both the Results and Discussion sections (Lines: 311-316, 322-331, 464-474). This information is not provided in the same manner by other assays. We have added an explanatory paragraph to point out the importance of these microscopy assays (Lines: 331-333).  

The FT-IR assay offers evidence of structural changes that occur during yeast cell lysis from a chemical perspective, which are explained in the Results and Discussion section (Lines: 490-496).

The β-glucan content of the obtained yeast cell walls highlights not only the purity of this derivative, but also the bioactive value contribution due to the presence of this valuable polysaccharide. We have rephrased Lines: 477-480, to emphasize this idea.

We have added some more comparisons of our results with literature data, regarding the protein content and the antioxidant activity of the obtained spent brewer’s yeast extracts (Lines: 434-447, 500-510). To the best of our knowledge, the studies on ultrasonication of spent brewer’s yeast are still rather scarce. The results and their explanations were correlated with literature data which is available on the topic at the moment, given the degree of novelty regarding this particular method for processing this still underutilized by-product (Lines: 416-421 provide information on the influence of duty cycle and sample concentration on the protein content, 427-434, 454-463 describe the impact of temperature and duty cycle on the released biocompounds, at lines 482-485 the β-glucan value is compared to those from literature data, at lines 278-299 the FT-IR spectrum bands are described with respect to literature values, at lines 464-474 a comparison of the microscopy assay with images found in literature is provided).

We have also revised the Materials and Methods section in order to achieve a more summarized version of this section (Lines: 128-132, 154-161, 183-184, 191-199).

Reviewer 2 Report

Comments and Suggestions for Authors

The current study deals with the valorization of spent brewer’s yeast bioactive components via an optimized ultrasonication process. The scope of the study is clearly stated, and statistical analysis of the results has been conducted. The M&M part is very descriptive and detailed enough for the reproducibility of the experiments. The results are clearly presented and compared and discussed with the ones of other authors in the literature. Generally, the manuscript has a nice flow and is easy to follow and understand. The findings of the study are interesting. I would suggest:

-          A minor revision in English (there are some “leftovers” from a previous version of the manuscript, use of between instead of among etc).

-          Pg.4 , ln 158. The referenced paragraph should be corrected (2.2.5 to 2.2.7).

-          Regarding the abstract, lns 21-22 should be rewritten since as “ultrasonication parameters” could be mostly characterized the ones we choose for the process and are related to the sonication bath itself (i.e. time of treatment, frequency, etc).

Comments on the Quality of English Language

A minor revision in English is suggested (there are some “leftovers” from a previous version of the manuscript, use of between instead of among etc).

Author Response

Dear Reviewer,

We appreciate your efforts to review our manuscript and we are grateful for your valuable comments, which we hopefully managed to materialize in an improved version of our work.

Hence, we have revised the manuscript according to your comments and suggestions. The changes we have made are detailed below. Please note that when the position of the modification made within the text is specified, reference is made to the line number in the revised manuscript saved with the track changes option.

Thank you for the appreciation of our work.

Comment 1: Pg.4, ln 158. The referenced paragraph should be corrected (2.2.5 to 2.2.7).

Authors response to Comment 2: We have corrected the referenced paragraph (Line: 173)

Comment 2: Regarding the abstract, lines 21-22 should be rewritten since as “ultrasonication parameters” could be mostly characterized the ones we choose for the process and are related to the sonication bath itself (i.e., time of treatment, frequency, etc.).

Authors response to Comment 3: We have rewritten this phrase in the abstract. (Line: 21)

Comment 3: A minor revision in English (there are some “leftovers” from a previous version of the manuscript, use of between instead of among etc).

Authors response to Comment 1: We have revised the English grammar and typos errors of the manuscript.

Reviewer 3 Report

Comments and Suggestions for Authors

What is the meanin g of Valorization?

Line …brewery industry with elevated bioactive potential.

Abstract should include key data and conclusion.

Line 101-102, Ultrasonic intensity density should be mentioned.

Line 110, 171, 182, please convert the unit rcf into g.

Line 120, After 30 min. of dark incubation at room temperature?

Please turn to a native speaker of English to improve the language, grammar and punctuation errors should be checked one by one.

Introduction should mention rationales of application of ultrasound to improve in the study (Enhancing the taste of raw soy sauce using low intensity ultrasound treatment during moromi fermentation. Food Chemistry, 2019, 298, 124928.; Ougan juice debittering using ultrasound-aided enzymatic hydrolysis: Impacts on aroma and taste. Food Chemistry, 2021, 345, 128767.)  

Comments on the Quality of English Language

What is the meanin g of Valorization?

Line …brewery industry with elevated bioactive potential.

Abstract should include key data and conclusion.

Line 101-102, Ultrasonic intensity density should be mentioned.

Line 110, 171, 182, please convert the unit rcf into g.

Line 120, After 30 min. of dark incubation at room temperature?

Please turn to a native speaker of English to improve the language, grammar and punctuation errors should be checked one by one.

Introduction should mention rationales of application of ultrasound to improve in the study (Enhancing the taste of raw soy sauce using low intensity ultrasound treatment during moromi fermentation. Food Chemistry, 2019, 298, 124928.; Ougan juice debittering using ultrasound-aided enzymatic hydrolysis: Impacts on aroma and taste. Food Chemistry, 2021, 345, 128767.)  

Author Response

Dear Reviewer,

We appreciate your efforts to review our manuscript and we are grateful for your valuable comments, which we hopefully managed to materialize in an improved version of our work.

Hence, we have revised the manuscript according to your comments and suggestions. The changes we have made are detailed below. Please note that when the position of the modification made within the text is specified, reference is made to the line number in the revised manuscript saved with the track changes option.

Comment 1: What is the meaning of Valorization?

Author response to Comment 1: Valorization refers to the aim of exploiting the spent brewer’s yeast biomass in a way that would enhance its value. We thought it was a suggestive term which encompasses the aim of this work.

Comment 2: Line …brewery industry with elevated bioactive potential.

Author response to Comment 2: We have rephrased, we hope it is clearer now (Line: 19)

Comment 3: Abstract should include key data and conclusion.

Author response to Comment 3: We have included key data and conclusions in the abstract (Line: 29-33).

Comment 4: Line 101-102, Ultrasonic intensity density should be mentioned.

Author response to Comment 4: We have stated out the ultrasonication energy density (Line: 116).

Comment 5: Line 101-102, Line 110, 171, 182, please convert the unit rcf into g.

Author response to Comment 5: We have changed the rcf unit to g (Lines: 122, 186, 199)

Comment 6: Line 120, After 30 min. of dark incubation at room temperature?

Author response to Comment 6: We have rephrased, since the described protocol refers to samples being kept at room temperature, and incubation was indeed not the precise term (Line: Line: 133).

Comment 7: Please turn to a native speaker of English to improve the language, grammar and punctuation errors should be checked one by one.

Author response to Comment 7: We have revised the English grammar and typos errors of the manuscript.

Comment 8: Introduction should mention rationales of application of ultrasound to improve in the study (Enhancing the taste of raw soy sauce using low intensity ultrasound treatment during moromi fermentation. Food Chemistry, 2019, 298, 124928.; Ougan juice debittering using ultrasound-aided enzymatic hydrolysis: Impacts on aroma and taste. Food Chemistry, 2021, 345, 128767.)

Author response to Comment 8: We have mentioned these compelling applications of this process, regarding the enhancement of taste in ultrasonicated products. (Lines: 63-69)

Round 2

Reviewer 1 Report

Comments and Suggestions for Authors

Check tables number at line 366 and 401

Reviewer 3 Report

Comments and Suggestions for Authors

The authors have revised the manuscript according to reviewers' comments, it can be accepted.